# Interaction of Arylidenechromanone/Flavanone Derivatives with Biological Macromolecules Studied as Human Serum Albumin Binding, Cytotoxic Effect, Biocompatibility Towards Red Blood Cells

**DOI:** 10.3390/molecules23123172

**Published:** 2018-12-01

**Authors:** Angelika A. Adamus-Grabicka, Magdalena Markowicz-Piasecka, Michał B. Ponczek, Joachim Kusz, Magdalena Małecka, Urszula Krajewska, Elzbieta Budzisz

**Affiliations:** 1Department of Cosmetic Raw Materials Chemistry, Faculty of Pharmacy, Medical University of Lodz, ul Muszynskiego 1, 90-151 Lodz, Poland; angelika.adamus@umed.lodz.pl; 2Laboratory of Bioanalysis, Department of Pharmaceutical Chemistry, Drug Analysis and Radiopharmacy, Medical University of Lodz, Muszyńskiego1, 90-151 Lodz, Poland; magdalena.markowicz@umed.lodz.pl; 3Department of General Biochemistry, Faculty of Biology and Environmental Protection, University of Lodz, Pomorska 141/143, 90-236 Lodz, Poland; michal.ponczek@biol.uni.lodz.pl; 4Institute of Physics, University of Silesia, Uniwersytecka 4, 40-007 Katowice, Poland; joachim.kusz@us.edu.pl; 5Department of Physical Chemistry, Theoretical and Structural Chemistry Group, Faculty of Chemistry, University of Lodz, Pomorska 163/165, 90-236 Lodz, Poland; magdalena.malecka@chemia.uni.lodz.pl; 6Department of Pharmaceutical Biochemistry and Molecular Diagnostics, Faculty of Pharmacy, Medical University of Lodz, Muszynskiego 1, 90-151 Lodz, Poland; urszula.krajewska@umed.lodz.pl

**Keywords:** synthesis, crystal structure, cytotoxic effect, benzoflavanone/chromanone derivatives, erythrotoxicity

## Abstract

The aim of this study was to determine the cytotoxic effect of 3-arylidenechromanone (**1**) and 3-arylideneflavanone (**2**) on HL-60 and NALM-6 cell lines (two human leukemia cell lines) and a WM-115 melanoma cell line. Both compounds exhibited high cytotoxic activity with higher cytotoxicity exerted by compound **2**, for which IC_50_ values below 10 µM were found for each cell line. For compound **1**, the IC_50_ values were higher than 10 µM for HL-60 and WM-115 cell lines, but IC_50_ < 10 µM was found for the NALM-6 cell line. Both compounds, at the concentrations close to IC_50_ (concentration range: 5–24 µM/L for compound **1** and 6–10 µM/L for compound **2**), are not toxic towards red blood cells. The synthesized compounds were characterized using spectroscopic methods ^1^H- and ^13^C-NMR, IR, MS, elemental analysis, and X-ray diffraction. The lipophilicity of both synthesized compounds was determined using an RP-TLC method and the log*P* values found were compared with the theoretical ones taken from the Molinspiration Cheminformatics (miLog*P*) software package. The mode of binding of both compounds to human serum albumin was assessed using molecular docking methods.

## 1. Introduction

Flavonoids constitute a large group of natural and synthetic polyphenolic compounds with a wide range of antioxidant, anti-allergic, anti-inflammatory, anti-microbial, anti-coagulant, anti-cholesterol, or anti-cancer activities. These properties often depend on unknown structure-related interactions with nucleic acids and proteins [1,2]. Due to their antioxidant properties, flavonoids interfere with the initiation, promotion, and progression of cancer. Flavonoids modulate the activities of various enzymes and receptors involved in several signal transduction pathways, thus affecting cell proliferation, apoptosis, as well as the processes of inflammation, angiogenesis, and metastasis [3]. Due to their multidirectional mechanisms of action, flavonoids are promising candidates for novel anti-cancer agents and are being extensively investigated for the treatment of neoplastic diseases [3,4,5]. For instance, it was found that, in vivo, quercetin exerts a chemo-preventive effect on prostate cancer (reduces the cell viability) by downregulating the activity of pro-proliferative and anti-apoptotic proteins [6]. Additionally, synthetic derivatives of naturally occurring flavonoids (e.g., derivatives of chroman-4-one and tiochromane-4-one [7]) are studied for their anti-cancer properties. 

3-arylidenechroman-4-ones, a group of naturally occurring homoisoflavones, can be obtained by the condensation of aryl aldehydes with chroman-4-ones (chromanones). It was first synthesized by Robinson and co-workers in the early twenties by condensation of chromanon with the corresponding aryl aldehyde using alcoholic potassium hydroxide as a catalyst [8]. Other catalysts have also been proposed. In 1979, Levai and Schag synthesized *E*-3-arylidenechroman-4-ones using piperidine as a catalyst, while piperidine has also been successfully used as a catalyst for the synthesis of 3-benzylidenechroman-4-ones and 3-benzylideneflavanones [9,10]. In recent years, Kupcewicz and co-authors [11], as well as our group [12], have investigated the cytotoxicity of *E* and *Z* isomers of 3-arylideneflavanone derivatives. Those compounds exhibit high cytotoxicity with IC_50_ < 10 μM. However, no significant difference in cytotoxicity of *E* and *Z* isomers was observed. In this study the structure of (*E*)-3-(4-*N*,*N*-diethylaminobenzylidene)chroman-4-one (**1**) and (*E*)-3-(4-*N*,*N*-diethylaminobenzylidene)-2-phenylchroman-4-one (**2**) was elucidated using single crystal X-ray diffraction (XRD), MS, and NMR spectroscopy. Despite being first synthesized in the 1970s [9,10], compound **1** was presented as an American patent in 2008 [13] and compound **2** was synthesized in 1993 by Pijewska [14], but their cytotoxic activity has never been studied. The cytotoxicity of 3-arylidenechromanone and flavanone substituted with *p*-aminodiethyl on C3 atom in benzene ring has also been investigated. In addition, the lipophilicity (miLog*P*) of **1** and **2** and their activity against red blood cells were determined. The mode of binding to human serum albumin (HSA, a protein important in binding and transport of various drugs [15]) has also been elucidated by molecular docking methods. Computational docking of ligand binding to target proteins is often performed as part of Structure-Based Virtual Screening used in drug discovery and this approach enables molecular interactions with macromolecules to be predicted [16]. We also assess the lipophilicity and their influence on the various biological properties. 

## 2. Results and Discussion 

### 2.1. Chemistry

The cytotoxic effects of *E* and *Z*-isomers of benzylideneflavanone are compared in a previous paper [11]. The current paper compares the cytotoxicity of chromanone and flavanone derivatives. (*E*)-3-(4-*N*,*N*-diethylaminobenzylidene)chroman-4-one (**1**) and (*E*)-3-(4-*N*,*N*-diethylaminobenzylidene)-2-phenylchroman-4-one (**2**) synthesized, as described previously [14] (Scheme 1).

Both compounds were fully characterized using IR, ^1^H-NMR, ^13^C-NMR, and MS spectroscopy and elemental analysis. 

The *ν* (C=O) band for compound **1** was observed at 1663 cm^−1^, and bands typical for aromatic rings were noted at 1600, 1584, and 1524 cm^−1^. An IR spectrum of **2** showed a band at 1654 cm^−1^ and typical bands for the aromatic rings at 1603, 1559, and 1524 cm^−1^. The ^1^H-NMR spectrum for compound **1** revealed the presence of a triplet from -CH_3_ protons at 1.12 ppm and a quartet around 3.42 ppm corresponding to the CH_2_ protons (see Appendix A). A singlet for =C-H was observed at δ 5.45 ppm. At δ 6.75 ppm, a doublet for the C2-H protons was observed. The characteristic signals δ 7.02–7.85 ppm correspond to the aromatic protons. The ^1^H-NMR spectrum for compound **2** revealed the presence of a triplet from -CH_3_ protons at 1.12 ppm (see Appendix A). A quartet corresponding to the CH_2_ protons was observed at δ 3.36 ppm. At δ 6.70 ppm a singlet for C2-H protons was noted. The signals at δ 6.81–7.79 ppm correspond to the aromatic protons. A singlet for =C-H was observed at δ 7.94 ppm. In the ^13^C-NMR spectrum for compound **1**, the characteristic CH_3_ group signal was found at δ 12.9 ppm, while the -CH_2_ group was observed at 44.3 ppm (see Appendix A). The C2-H carbon signal was observed at δ 78.1 ppm. At δ 181.0 ppm, a signal for C=O was observed. In addition the ^13^C-NMR spectrum showed the presence of CH_arom_ and C_arom_ signals, which were expected for the postulated structure of (*E*)-3-(4-*N*,*N*-diethylaminobenzylidene)chroman-4-one. For compound **2**, the ^13^C-NMR spectrum showed the presence of signals located at δ 12.9 ppm, attributed to a CH_3_ group and at -CH_2_ group at δ 44.3 ppm (see Appendix A). In the ^13^C-NMR spectrum, a -C=O related resonance was found at δ 181.2 ppm, those for CH_arom_ and C_arom_ were also found. The mass spectrum revealed a band in **1** at *m*/*z* = 308.2, indicating ([M + H]^+^), and in **2**, at *m*/*z* = 384, indicating the presence of an [M + H] ion.

### 2.2. Molecular Structures

The molecular structure of compound **1** and **2** is presented in (Figure 1). A molecule of **1** consists of two condensed rings (benzene and pyran) with 4-*N*,*N*-diethyloaminobenzylidene substituent in position 3 of the benzopyran ring. It crystalizes in a triclinic system in a P$\bar{1}$ space group with two molecules in the asymmetric unit. The two independent molecules demonstrate different geometries in the arrangement of the 4-*N*,*N*-diethylaminobenzylideno moiety with respect to the main benzopyran skeleton. In molecule A, it is nearly planar with dihedral angles C3-C11-C12-C13 −8.32(1) and C3-C11-C12-C17 172.80 (1)°, while in molecule B, it is rotated by about 35° with the dihedral angles C53-C61-C62-C63 of −35.41(1)° and C53-C61-C62-C67 of 148.31 (1). Moreover, the pyran rings slightly vary in conformation. In molecule A, the pyran ring adopts an envelope conformation with puckering ring parameters [11] Q = 0.300(2)Å, ϕ = 57.0(3), θ = 62.5(2)°, and the asymmetry parameter [12] ΔC_s_(C2) = 2.75(1)°. The corresponding parameters for the pyran ring in the molecule B are: Q = 0.451(2)Å, ϕ = 44.9(3)°, θ = 64.6(2)°, and the asymmetry parameter ΔC_2_(O51-C52) = 14.0(1)°, which indicates a screw-boat conformation. Valkonen [17] notes that, in the structure of (*E*)-3-(4-(dimethylamino)benzylidene)-2,3-dihydro-4*H*-chromen-4-one, there are also two independent molecules crystalizing in an asymmetric unit of the Pc group. The benzopyran skeleton and the 4-dimethylaminobenzylideno moiety are not planar with dihedral angles of 24.96(2)° and 49.53(2)° for molecules A and B, respectively. For the C6-C10-C2-C12/C9 and C22-C2)-C19-C21/C31, the corresponding torsion angles of 32.65(2)°/149.33(2)°, −29.42(2)°, and 156.30(2)° were reported. However, the pyran rings adopt a similar conformation: An envelope conformation with the puckering ring parameters [18] Q = 0.349(2)Å, ϕ = 309.1(2), θ = 58.6(2)°, the asymmetry parameter [19] ΔC_s_(C4) = 7.30(1)° for O1/C5/C6/C3/C4/C1 ring, a screw boat conformation with Q = 0.405(2)Å, ϕ = 143.0(2), θ = 113.5(2)°, and the asymmetry parameter [20] ΔC_2_(C24-C29) = 4.91(1)° for the O4/C28/C22/C29/C24/C26 ring

The crystal packing of **1** is dominated by C-H…O and C-H…π interactions (Figure 2). The molecules A and B are linked by C8-H8…O2 (−1 + x, y, z) and C67-H67…O51 (−1 + x, y, z) hydrogen bonds, respectively, forming two ribbons, which are further coupled by C56-H56…O2 (x, 1y, z) hydrogen bond that produces a layer. Other C52-H52…O51 (1 − x, 1 − y, 1 − z) interactions are responsible for creating the 3-D network of hydrogen bonds connecting two layers. In both molecules, the intramolecular hydrogen bonds C11-H11…O2 and C61-H61…O52 are observed (Table 1).

Compound **2** crystalizes in a monoclinic system in the P2_1_/n space group. The main skeleton of the molecule consists of fused benzene and pyran rings, the latter bears a phenyl ring at position 2 and a 4-*N,N*-diethyloaminobenzylideno substituent at position 3. The C-2 phenyl ring is perpendicular, while the aminobenzylidene substituent is nearly co-planar to the pyran ring. The corresponding dihedral angles are: C3-C2-C22-C27 −13.0(2)° and C3-C11-C12-C13 19.2(3)°, while the dihedral angle between the best planes of those phenyl rings is 83.7(2)°. In molecule **2**, the pyran ring adopts a distorted half-boat conformation with the puckering ring parameters [21] Q = 0.411(2)Å, ϕ = 220.6(3)°, θ = 119.2(2)°, and the asymmetry parameter [22] ΔC_2_(O1-C2) = 7.80(2)°.

In comparison to previously investigated similar molecules [11], the appropriate torsion angles are: −16.4(2)° and 32.5(2)° with a dihedral angle observed between the best planes of phenyl rings 52.8(1)°. The pyran ring also adopts a nearly half boat conformation with puckering parameters [20] Q = 0.341(2)Å, ϕ = 226.1(3)°, and θ = 120.3(2)°.

In the crystal packing of **2**, hydrogen bonded dimers are observed, and these are stabilized by C16-H16…O2 interactions (1 − x, 1 − y, −z) (Figure 3). The intramolecular C11-H11…O2 hydrogen bond is also observed similar to compound **1**. The crystal packing is also enhanced by C-H…π interactions (Table 2).

### 2.3. Biological Assays

#### 2.3.1. Cytotoxic Activity

The cytotoxicity of the synthesized compounds was evaluated in vitro by the MTT assay using three human cancer cell lines HL-60, NALM-6, and WM-115. The results, expressed as inhibitory concentration (IC_50_), are shown in Table 3. Compound **2** exhibited high cytotoxic activity with IC_50_ = 6.45 ± 0.69 µM, which is nine times lower than for 3-benzylideneflavanone and 154 times lower than for 4-chromanone, the reference compounds used in the WM-115 cell line. For the other cell lines, IC_50_ values were also satisfactory. In the NALM-6 cell culture, compound **1** showed an IC_50_ value almost 78 times lower than for 4-chromanone and more than three-fold lower than for 3-benzylideneflavanone. Perjesi and coworkers [23] also reported cytotoxicity of a series of 3-benzylidene-4-chromanones towards several cell lines. The authors found that most of the compounds inhibited the growth of 50% of HL-60 cells at the concentrations between 3–28 µM with 3-(3-bromobenzylidene)-2,3-dihydro-1-benzopyran-4-one as the most active. In the current study we reveal that both synthesized compounds (**1** and **2**) exhibit good anti-proliferative properties at low micromolar range which make them promising candidates for further in vitro and in vivo studies.

#### 2.3.2. Rbcs Lysis Assay

In studies on the integrity of the erythrocyte membrane, statistically significant results in comparison with saline control were obtained for compound **2** at the highest concentrations tested (9 and 10 μmol/L) (Figure 4), which implies that compounds at these concentrations contributed to red blood cell breakdown. However, the values of hemolysis did not exceed 10%, which is considered to be a clinically important threshold of cytotoxicity [24].

In contrast, compound **1** did not lead to any significant increase in the hemolysis rate, therefore, it might be concluded that both examined compounds do not show adverse effects on the integrity of the red blood cells (RBCs) membrane over the entire concentration range.

#### 2.3.3. RBCs Morphology

When a drug molecule is administered intravenously a series of interactions with blood cells might be initiated. The effects of **1** and **2** on the morphology of erythrocytes were visualized by microscope studies (Figure 5).

Both tested compounds **1** and **2**, contributed to the formation of echinocytes. Echinocytes accounted for the major part of erythrocytes when 24 µmol/L compound **1** was applied. As presented in Figure 5, compound **2** led to the formation of echinocytes over the entire concentration range, however, the percentage of this erythrocyte form differed according to the concentration of **2**. The transformation of biconcave erythrocytes to echinocytes occurs naturally in blood vessels, and could be also seen in control samples. It has been proved that echinocytosis is a reversible transformation caused by various chemical and physical factors including increased ion strength, alkaline pH, and decreased in adenosine triphosphate (ATP) level [25].

### 2.4. Computational Studies

When drugs circulate in blood, they interact with biological macromolecules such as plasma proteins; these albumins constitute the majority and are important in the pharmacokinetics of medicines. For derivatives **1** and **2**, tenfold blind experiments of docking to HSA were performed to establish locations of their binding sites and the strength of the hypothetical interactions. The compounds were found to bind through hydrophobic interactions, **1** in four different places (near the cleft site, at the IB and FA1 sites, above the IIA—drug site and some additional atypical place) with average ΔG° −7.5 kcal/mol, whereas **2**, only in the IB FA1 site with average ΔG° −9.5 kcal/mol (Figure 6A,B). This single binding most likely resulted from the presence of an additional phenyl ring at the C2 position. The π…π stacking was an important interaction between phenyl rings of the docked compounds and the aromatic side chains of tyrosine and phenylalanine. In the IB FA1 site, the aromatic rings of Tyr138 and 4-diethyloaminobenzylidene substituent were parallel and were only slightly offset from each other with a distance between 4.4 and 4.5 Å. The phenyl ring in position 2 was located between the phenyl rings of Tyr138 and Phe125. The pyran ring was located between the phenyl ring of Phe125 and the aliphatic chain of Lys137 (Figure 6C,D). In contrast, compound 1 was located in the IB FA1 site inversely and showed less potent hydrophobic interactions. The rings of the 4-*N*,*N*-diethyloaminobenzylidene substituent and Tyr138 were arranged in a parallel orientation, but were much more offset. The pyran ring was perpendicular to the phenyl ring of Tyr140 with a distance greater than 5 Å. His247 and Tyr30 were isolated with the distance greater than 20 Å. For compound **1**, mainly hydrophobic interactions with the aliphatic amino acid residues were observed in the other binding sites, and in the case of the cleft site, an oblique interaction between the pyran ring with the Tyr452 phenyl ring could be possible within a distance of 3.6 Å. In one atypical place, a similar interaction with the imidazole ring of His146 could take place within a distance of 3.3 Å. No specific hydrogen bonds were detected for either compounds.

To predict other properties of compounds **1** and **2**, the Molinspiration web tool was used and the drug descriptors were counted. For the Molinspiration GPCR ligand, Ion channel modulator, Kinase inhibitor, Nuclear receptor ligand, enzyme inhibitor and protease inhibitor bioactivity the scores were < zero or near zero, indicating that the compounds were not typical inhibitors or agonists of the listed types of proteins. Ligand efficiencies (LE) for derivatives calculated from computationally estimated ΔG° and known number of heavy atoms, theoretical log*P* and log*P*/LE (LELP), divided by the predicted number of binding sites were different for two compounds, but were still within the range of drug-likeliness (Table 4).

The experimental lipophilicity of the synthesized compounds was determined using RP-TLC: log*P* values 3.43 was obtained for compound **1** and 5.69 for compound **2**. Theoretical values of lipophilicity were calculated using Molinspiration Cheminformatics (miLog*P*):milog*P* 4.47 was obtained for **1** and 6.05 for **2**, i.e., similar to those obtained experimentally. Greater values of experimental and theoretical lipophilicity were noted for **2** because of the presence of the phenyl ring in the C2 position. Thus, molecular docking and Molinspiration predictions suggest that **1** and **2** have greater hydrophobicity than **1**, but also indicate potential specificity influence of an additional phenyl ring.

## 3. Materials and Methods

The second compound **2** was synthesized according to Pijewska [14]. Compounds **1** and **2** were purified by crystallization from methanol. All solvents (methanol, ethanol, and toluene) used in this work were purchased from Sigma-Aldrich (St. Louis, MO, USA) and POCH (Gliwice, Poland) chemical companies and were used without further purification.

Melting points were determined on a Büchi Melting Point B-540 apparatus in the capillary mode and they were uncorrected. The infrared transmission spectra of the crystalline products were recorded using a Nexus Thermo Nicolet FT-IR spectrophotometer (Wien, Austria; Faculty of Chemistry, University of Lodz). The MS-ESI were measured on Varian 500, MS LC Ion Trap mass spectrometer (Santa Clara, CA, United States; Faculty of Chemistry, University of Lodz). Elemental analyses were performed in the Faculty of Chemistry (University of Lodz) using a Vario Micro Cube (Langenselbold, Germany) by Elemental analyzer. ^1^H- and ^13^C-NMR spectra were recorded on a Bruker Avance III 600 MHz (Karlsruhe, Germany). Both samples were dissolved in DMSO deuterated. Chemical shifts are given in ppm, coupling constants in Hz. Computational studies were used to predict the properties and future use of potential drugs. Molinspiration Cheminformatics (http://www.molinspiration.com/products.html) was used for the calculation of important molecular properties, molecular processing, bioactivity, and the high-quality depiction of new molecules. Molecular docking of the compounds to the HSA crystal structure 1E7I [21] was performed in AutodockVina 1.1.2 (http://vina.scripps.edu) [22], as previously described [12].

Lipophilicity was determined using an RP-TLC method with small amounts of organic solvents.

Assessment of erythrotoxicity of compounds **1** and **2** was made in Laboratory of Bioanalysis in the Department of Pharmaceutical Chemistry, Drug Analysis, and Radiopharmacy in the Medical University of Lodz. The erythrocyte morphology was evaluated using a phase-contrast microscope Opta-Tech (Warsaw, Poland) using the OptaView 7 programme. Biological material (blood) for research came from the Regional Blood Donation Center in Lodz.

### 3.1. Synthesis of (E)-3-(4-N,N-diethylaminobenzylidene)chroman-4-one (**1**)

A mechanically stirred mixture of 10 mmol of chroman-4-one, 10 mmol 4-*N*,*N*-diethylaminobenzaldehyde and five drops of piperidine was heated at 140 °C in an oil bath for 4 h. After cooling the reaction mixture was left for 24 h at room temperature. Next, methanol was added and the resulting was filtered and crystallized from methanol. Compound **1** was thus obtained as a light orange powder.

Yield: 33.5%. M.p. 124.9–125.4 °C. MS (ESI^+^): *m*/*z* 308. C_20_H_21_NO_2_ [M + H]^+^. IR (KBr) ν (cm^−1^): 3037 (C-H_aromat_), 2968, 2923, 2895, 2870 (C-H_aliph_), 1663 (C=O), 1600, 1584, 1524 (C=C), 1154 (C-O-C). ^1^H*-*NMR (600 MHz, DMSO-d6) δ (ppm): 1.12 (3H, *t*, *J*_H-H_—6 Hz, CH_3_), 3.42 (2H, *q*, *J*_H-H_—6 Hz, CH_2_), 5.45 (1H, *s*, CH), 6.75 (2H, *d*, *J*_H-H_—6 Hz, C2-H), 7.02 (1H, *d*, *J*_H-H_—12 Hz, C4-H_arom_), 7.09 (1H, *t*, *J*_H-H_—6 Hz, C6-H_arom_), 7.32 (1H, *d*, *J*_H-H_—6 Hz, C8-H_arom_), 7.54 (1H, *t*, *J*_H-H_—6 Hz, C7-H_arom_), 7.84 (1H, *d*, *J*_H-H_—6 Hz, C6′-H_arom_), 7.85 (1H, *d*, *J*_H-H_—6 Hz, C5′-H_arom_). ^13^C-NMR (600 MHz, DMSO-d6) δ (ppm): 12.9 (CH_2_**C**H_3_), 44.3 (**C**H_2_CH_3_), 68.4 (**C**-2), 111.6, 118.1, 120.8, 122.2, 122.4, 125.2 (CH_arom,_ =CH), 135.9, 133.7, 137.9, 149.2, 160.7 (C_arom_), and 181.0 (C=O). Anal. Calc. for C_20_H_21_NO_2_ (M = 307.386 g/mol) % C:78.14; and % H: 6.89. Found % C: 78.16; and % H: 7.06.

### 3.2. Synthesis of (E)-3-(4-N,N-diethylaminobenzylidene)-2-phenylchroman-4-one (**2**)

The compound **2** was synthesized according to Pijewska in 1993 [14]. A racemic mixture was obtained.

Yield: 34%. M.p. 170–171 °C (MeOH). MS (ESI^+^): *m*/*z* 384. C_26_H_25_NO_2_ [M + H]^+^. IR (KBr) ν (cm^−1^): 3056 (C-H_aromat_), 2965, 2920, 2895, 2863 (C-H_aliph_), 1654 (C=O), 1603, 1559, 1524 (C=C), 1150 (C-O-C). ^1^H-NMR (600 MHz, DMSO-d6) δ (ppm): 1.12 (3H, *t*, *J*_H-H_—6 Hz, CH_3_), 3.36 (2H, *q*, *J*_H-H_—6 Hz, CH_2_), 6.70 (1H, s, C2-H), 6.81–7.79 (13H, m, C-H_arom_), 7.94 (1H, s, =CH). ^13^C-NMR (600 MHz, DMSO-d6) δ (ppm): 12.9 (CH_2_**C**H_3_), 44.3 (**C**H_2_CH_3_), 78.1 (C2-H), 119.0, 120.2, 125.9, 127.2, 127.9, 129.1, 129.3, 136.2, 138.6, 140.3 (CH_arom,_ =CH), 122.3, 122.7, 122.7, 133.7, 149.6, 158.5 (C_arom_), 181.2 (C=O). Anal. Calc. for C_26_H_25_NO_2_ (M = 389.529 g/mol) % C: 81.43; % H: 6.57; % N: 3.65. Found % C: 81.57; % H: 6.78; and % N: 3.58.

### 3.3. Determination of Lipophilicity of Flavone Derivatives Using RP-TLC Method

The RP-TLC experiments were performed on TLC plates (5 × 10 cm) RP-18 F254S (Merck, Darmstadt, Germany). The synthesized compounds were dissolved in *N*,*N*-dimethylformamide DMF (2 mg/mL). High purity DMF 99.8% was obtained from Chempur (Piekary Slaskie, Poland). The solutions of each compound in DMF were spotted on plates. The spots were observed under UV light at λ = 254 nm. DMF-water solvent system was used as mobile phase. The composition of the solvent system changed from 50%:50% to 95%:5%. All experiments were performed at room temperature. The Log*P* parameter was calculated using the equation from the calibration curve.

### 3.4. X-ray Diffraction Experiment

The X-ray single crystal diffraction experiments for **1** and **2** were performed using an Agilent SuperNova diffractometer equipped with an Atlas detector at T = 100(2)K with monochromatic MoKα radiation (λ = 0.71073 Å). Multi-scan absorption correction was applied to all data [26]. All structures were solved by direct methods using SHELX and further refined on F^2^ using SHELX-2014/7 [27]. The position of the hydrogen atoms were calculated from the known geometry (C-H bond lengths at 0.93, 0.96, and 0.97 Å for aromatic CH, methylene CH_2_, and CH_3_ methyl atoms, respectively) and treated as riding, where isotropic thermal parameters of these hydrogen atoms were fixed as U_iso_(H) = 1.2 U_eq_(C) (for aromatic and methylene H atoms) or U_iso_(H) = 1.5 U_eq_(C) for (methyl H-atoms). Key experimental crystallographic data and refinement details of the studied compounds **1** and **2** are summarized in Table 5. To identify molecular geometries and hydrogen-bond patterns, PLATON and Mercury [28] were utilized. CCDC 1877734 (compound **2**) and 1877735 (compound **1**) contains the Appendix A for this paper. These data can be obtained free of charge via www.ccdc.cam.ac.uk/data_request/cif, or by emailing data_request@ccdc.cam.ac.uk, or by contacting The Cambridge Crystallographic Data Centre, 12, Union Road, Cambridge CB2 1EZ, UK; fax: +44-1223-336-033.

### 3.5. Cells Cultures and Cytotoxicity Assay by MTT

Cytotoxicity was tested against human skin melanoma cells (WM-115 ECACC, Salisbury, UK), and two human leukemia cell lines—promyelocytic leukemia (HL-60) and lymphoblastic leukemia (NALM-6). The leukemia cells were cultured in RPMI 1640 medium (Invitrogen, Grand Island, NY, USA) supplemented with 10% fetal bovine serum (FBS; Invitrogen, Grand Island, NY, USA) and gentamicin (25 µg/mL; KRKA, Slovenia). For melanoma WM-115 cells, Dulbecco`s minimal essential medium (DMEM; Invitrogen, Grand Island, NY, USA) 1640 was used. Cells were grown at 37 °C in a humidified atmosphere of 5% CO_2_ in air.

For all experiments, the studied compounds were dissolved in DMSO (Sigma-Aldrich, St. Louis, MO, USA) and were further diluted in a culture medium to obtain <0.1 % DMSO concentration. In each experiment control without and with 0.1% DMSO was performed.

The cytotoxicity of both compounds, and the reference compounds (4-chromanone and 3-benzylidene-flavanone) was determined by the 3-(4,5-dimethylthiazol-2-yl)-2,5-diphenyltetrazolium bromide (MTT; Sigma, St. Louis, MO, USA) assay, which measures the activity of cellular dehydrogenases [29]. Exponentially growing cells were seeded on 96-well plates (Nunc, Roskilde, Denmark). Subsequently, various concentrations of the studied compounds (freshly prepared in DMSO and diluted with complete culture medium) were added. All compounds were tested for their cytotoxicity at a final concentration of 10^−7^–10^−3^ M. After 46 h of incubation with the studied compounds, the cells were treated with the MTT reagent and incubation was continued for another two hours. The MTT formazan crystals were dissolved in 20% SDS (Sodium dodecyl sulphate, Sigma-Aldrich, St. Louis, USA) and 50% DMF (Sigma-Aldrich, St. Louis, MO, USA) at pH 4.7, and the absorbance was read at 570 nm on a multifunctional Victor ELISA-plate reader (Perkin Elmer, Turku, Finland). The IC_50_ values, the concentration of the test compound, was required to reduce the cell survival fraction to 50% of the controls, and were calculated from concentration response curves and was used as a measure of the sensitivity of the cells to a given treatment. As a control, cultured cells were grown in the absence of drugs. The data points represent the means of at least five to ten repeats ± S.D (standard deviation).

### 3.6. Red Blood Cells Lysis Assay

The studies on biological material were approved by the Bioethics Committee of the Medical University of Lodz, Poland (RNN/27/18/KE). The blood from healthy donors (Blood Donation Centre in Lodz) was collected in tubes containing a solution of potassium EDTA. RBCs were isolated by centrifugation (3000× *g*, 10 min) at 20 °C and washed three times with 0.9% saline. The studies were performed on four biological samples obtained from different blood donors.

The effects of (*E*)-3-(4-*N*,*N*-diethylaminobenzylidene)-2-phenylchroman-4-one **1** and (*E*)-3-(4-*N*,*N*-diethylaminobenzylidene)-2-phenylchroman-4-one **2** on RBCs was performed using the method described earlier [30,31]. Briefly, 2% RBCs suspension prepared in 0.9% saline was incubated at 37 °C with tested compounds added in a 10 µL volume for one hour. The percentage of solvent (methanol) did not exceed 1% of the final sample volume (1 mL). Afterwards, the samples were centrifuged at 1000× *g* for 10 min (20 °C), and the absorbance of the supernatant was measured at 550 nm wavelength (spectrophotometer (Cecil CE2021, London, England)). Samples containing saline solution were used to measure spontaneous hemolysis of RBCs, while samples with Triton X-100 ((2.0% *v*/*v*); Polish Chemical Reagents) constituted 100% of hemoglobin release. The results are presented as a percentage of released hemoglobin. The examined concentrations of tested compounds were chosen on the basis of obtained IC_50_ values determined for three human cancer cell lines: HL-60, NALM-6, and WM-115.

### 3.7. RBCs Morphology

A 2% erythrocyte suspension was incubated at 37 °C for 1 h with various concentrations of tested compounds. The morphology of the RBCs was evaluated using a phase contrast Opta-Tech inverted microscope, at 400 times magnification, equipped with image analysis software (OptaView 7).

### 3.8. Statistical Analysis

Statistical analysis was conducted with a commercially available package (Statistica 12.0, StatSoft). All results are presented as mean ± SD. The normality of the distribution of continuous variables was confirmed with the Shapiro-Wilk test. Paired t-tests were used for intergroup comparisons of normally distributed variables. The results were considered significant at *p*-values lower than 0.05.

## 4. Conclusions

This study presents the synthesis, chemical characterization, and biological activity of 3-benzylidenederivatives of chromanone and flavanone. The structure of synthesized compounds was confirmed using NMR, MS, IR, and single crystal X-ray diffraction (XRD) techniques. The anti-cancer properties of synthesized compounds **1** and **2** were evaluated using HL-60, NALM-6, and WM-115 cancer cell lines. Both compounds exerted a greater cytotoxic effect towards all cell lines than the reference 4-chromanone and 3-benzylideneflavanone. Compound **2** exerted the most profound cytotoxic effect towards WM-115 cell line with IC_50_ = 6.45 ± 0.69 µmol/L. Lipophilicity of both compounds was determined using an RP-TLC method. Due to the presence of a phenyl substituent in the C2 position of chromanone, the value of log*P* for compound **2** was higher than that for **1**. Both compounds did not affect the integrity of erythrocyte membrane over the examined concentration range and did not contribute to hemolysis. In addition, tested compounds did not induce the pathological changes in the morphology of erythrocytes since only dyscocytes and echinocytes were recognized. Thus, both compounds at the concentrations close to the determined IC_50_ values might be regarded as biocompatible towards red blood cells. Computational studies suggest remarkably different ability of the synthetized compounds to bind with HSA, and different specificity as far as the future use as drugs is considered. Taking into consideration the promising cytotoxicity properties, i.e., IC_50_ values in a few micromolar range and being harmless to the red blood cells, the arylidenechromanone/flavanone derivatives can be further evaluated as promising anticancer compounds.

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
