# Peer review of "Interaction of Arylidenechromanone/Flavanone Derivatives with Biological Macromolecules Studied as Human Serum Albumin Binding, Cytotoxic Effect, Biocompatibility Towards Red Blood Cells"

_molecules, 2018, doi:10.3390/molecules23123172_

Round 1
Reviewer 1 Report
The manuscript entitled
‘Interaction of arylidenechromanone/flavanone derivatives with biological macromolecules studied as human serum albumin binding, cytotoxic effect, biocompatibility towards red blood cells and their X-ray structures’
by Adamus-Grabicka, Markowicz-Piasecka, Ponczek, Kusz, Małecka, Krajewska, and Budzisz
reports a study on (E)-3-(4-N,N-diethylaminobenzylidene)chroman-4-one (1) and (E)-3-(4-N,N-diethylaminobenzylidene)-2-phenylchroman-4-one (2). In particular, they report the X-ray diffraction analysis data of the two compounds, and their lipophilicity. Biological essays showed differences in cytotoxicity of 1 and 2 against HL-60, NALM-6, and WM-115 cancer cell lines. The cytotoxicity activity was also compared with that of 4-chromanone and 3-benzylideneflavanone. Studies on RBCs lysis and morphology were also reported.
A docking study on the binding of 1 and 2 with human serum albumin was carried out.
In my opinion, the manuscript must be improved by addressing to the following main points:
· X-ray crystallographic data for 1 and 2 must be deposited in a database, e.g. CCDC, as indicated in instructions for authors.
· Authors must resize them Figures 2 and 3, that are unreadable.
· •The X-ray crystallographic data for the N,N-dimethyl analogous of compound 1 have been reported in ‘Structural Chemistry’ 2012, Volume 23, pp 209–217 and deposited with number: CCDC-746345. The authors might make a comparison.
· Table 3, first row: it should be indicated that reported values are IC50
· The N,N-dimethyl analogous of compound 1 has been reported in many papers. In European Journal of Medicinal Chemistry 2008 vol. 43 p. 839 – 845, Perjesi, Pal et al. reported the biological activity on a series of cancer cell lines, including HL-60. The authors might try a comparison.
· Row 179: Table 4 must be corrected into Table 3.
· Materials and Methods: 13C NMR data for compound 1 must be added.
· Materials and Methods: NMR data for compound 2 have been reported in CDCl3. In this manuscript, authors report 1H NMR data in DMSO-d6; they must add 13C NMR data in DMSO-d6.
Furthermore, the language must be re-checked and some language errors must be corrected.
On the whole, the manuscript can be published only after the above indicated corrections/modifications.
Reviewer 2 Report
This manuscript reports results on the synthesis of an arylidenechromanone (1) and an arylidene flavone (2) and a comparative study of their citotoxic activity on two human leukemia cell lines and a melanoma cell line.
This constitutes preliminary interesting results, but the article has not yet reached the state and the standars that may justify its publication in Molecules. Results on only two products are reported. A more robust article including more products is required for getting conclusive results.
In addition, major and minor formal considerations should be taken into account, as follows:
1. A shorter, informative title is desirable. The reference to X ray structures should be removed, taking into account that, in my opinion, this is a side issue in the context of the article.
2. A new, shorter version of the article is desirable, were typewriting mistakes and verbosity should be avoided. Moreover, the English wording should be revised by an specialist (please see representative examples along the attached version of the manuscript). I ask the authors to consider one of their previous articles [Bioorganic & Medicinal Chemistry Letters 23 (2013) 4102–4106] as a good model for this.
3. Subsection 2.1 should be improved as follows:
a. Figure 1 should be replaced by a Scheme depicting the chemistry involved in the preparation of targets 1 and 2.
b. A stereogenic center is present in flavone 2. I presume that a racemic mixture has been obtained. If so, are both enantiomers isolable?. This is important, in order to establish which one is active.
c. Have the authors considered to attempt a stereoselective version of the reaction leading to compound 2?
d. The authors comment the 1H NMR spectrum of compound 1 only. Why not the same for compound 2?
e. The graphs of the NMR spectra should be included in the supporting information.
f. Why 13C NMR spectra were not recorded?
g. Why the m/z values of the low resolution mass spectra include decimals? Please explain.
4. Subsection 2.2.
Why inclusion of this section in the manuscript? It consists of side, unnecessary information in the context of the article. Please, consider to remove it or, altertatively, include in the manuscript only a short comment similar to that in the above mentioned article by the authors [Bioorganic & Medicinal Chemistry Letters 23 (2013) 4102–4106], and translate the rest to the supporting information document. Please, justify the option selected.
Additional comments are included in the revised copy of the manuscript.

Author Response
Response to the Reviewer
General Response: According to the Editor’s suggestion the English was corrected, and the conclusion was rewritten.
Reviewer #2
1. Comment: A shorter, informative title is desirable. The reference to X ray structures should be removed, taking into account that, in my opinion, this is a side issue in the context of the article.
Response: We shortened the title according to the Reviewer suggestion. The present title is: “Interaction of arylidenechromanone/flavanone derivatives with biological macromolecules studied as human serum albumin binding, cytotoxic effect, biocompatibility towards red blood cells”
2. A new, shorter version of the article is desirable, were typewriting mistakes and verbosity should be avoided. Moreover, the English wording should be revised by an specialist (please see representative examples along the attached version of the manuscript). I ask the authors to consider one of their previous articles [Bioorganic & Medicinal Chemistry Letters 23 (2013) 4102–4106] as a good model for this.
Response: In the manuscript introduction was shortened. We have removed the mistakes indicated by the Reviewer, and corrected typographical errors throughout the manuscript.
3. Subsection 2.1 should be improved as follows:
a. Figure 1 should be replaced by a Scheme depicting the chemistry involved in the preparation of targets 1 and 2.
Response: Figure 1 was replaced by a Scheme 1. We have presented a method for synthesis of compounds 1 and 2.
b. A stereogenic center is present in flavone 2. I presume that a racemic mixture has been obtained. If so, are both enantiomers isolable?. This is important, in order to establish which one is active.
Response: In the crystal structures (solid state) of both compounds we obtained racemic mixture, because of centrosymmetric Space Groups (P-1 for compound 1 and P21/n for compound 2). The racemic substrate has been used in order to obtain compound 2.
Given the time limit for resubmission of revised manuscript we are unable to develop the method to isolate both enantiomers. But certainly it is highly important to determine which the isomer or the racemic mixture exert beneficial anti-proliferative activity.
c. Have the authors considered to attempt a stereoselective version of the reaction leading to compound 2?
Response: Yes, we have considered to conduct a stereoselective reaction in order to obtain one of the enantiomer, but this research is a subject of our next/future project.
d. The authors comment the 1H NMR spectrum of compound 1 only. Why not the same for compound 2?
Response: The spectrum description for compound 2 has been completed.
e. The graphs of the NMR spectra should be included in the supporting information.
Response: NMR spectra were included in the supplementary materials.
f. Why 13C NMR spectra were not recorded?
Response: In present version we added the 13C NMR data.
g. Why the m/z values of the low resolution mass spectra include decimals? Please explain.
Response: The m/z values was corrected.
4. Comment: Subsection 2.2.
Why inclusion of this section in the manuscript? It consists of side, unnecessary information in the context of the article. Please, consider to remove it or, altertatively, include in the manuscript only a short comment similar to that in the above mentioned article by the authors [Bioorganic & Medicinal Chemistry Letters 23 (2013) 4102–4106], and translate the rest to the supporting information document. Please, justify the option selected.
Response: This section has been removed and transfer to section Computational Studies.
5. Additional comments are included in the revised copy of the manuscript.
Response: The comments included in the manuscript have been taken into account, and corrected

Reviewer 3 Report
In this paper the author has conducted the comprehensive study of two compounds the: 3-arylidenechromanone (1) and 3-arylideneflavanone (2). The structures of two compounds were characterized by 1H-NMR, IR, MS, elemental analysis and their single crystal structures were analyzed by X-ray diffraction. Both compounds exhibit high cytotoxic activity against HL-60, WM-115 and NALM-6 cell line by MTT. At the same time, both compounds are considered biocompatible towards red blood cells at the concentrations corresponding to cytotoxic properties by red blood cells lysis assay and RBCs morphology image analysis. Molecular docking and Molinspiration predictions were also discussed. In addition, the author showed the binding of two compounds into human serum albumin using molecular docking methods.
Author Response
Authors thank reviewer for useful comments.
Round 2
Reviewer 1 Report
The manuscript, after revision, has been strongly improved and it can be accepted for publication.
Author Response
Reviewer #1:
The Authors thank to the Reviewer for careful reading of the text and all comments. The English language was corrected by a native speaker, and we also declare that if it is necessary we can include the certificate.
Reviewer 2 Report
The authors have reasonably taken into account my comments and objections. Accordingly, I can accept its publication in this journal after the following minor changes:
Page3, Scheme 1.- Why O atoms in red colour and N atom in blue color?
Page 9, lines 336-337.- The statement "6.70 (1H, s, JH-H – 6 Hz, C2-H)" is wrong. A J value es provided for a singlet. Please, correct.
Supporting Information File.
1. The title "Interaction of arylidenechromanone/flavanone derivatives with biological macromolecules studied as human serum albumin binding, cytotoxic effect, biocompatibility towards red blood cells and their X-ray structures" is not the same as for the manuscript. Please remove "and their X-ray structures".
2. The graphs of the 1H NMR and 13C NMR spectra should be included in the supporting information file
Author Response
Reviewer #2
1. Comment: Why O atoms in red colour and N atom in blue color?
Response: The scheme of compounds had been drawn in Accelrys Draw 4.1 programme. Heteroatoms are always marked with colours: oxygen in red, nitrogen in blue, sulfur in yellow, chlorine in purple. The colours of the atoms in the compond 1 and 2 were changed to black.
2. Comment: The statement "6.70 (1H, s, JH-H – 6 Hz, C2-H)" is wrong. A J value es provided for a singlet. Please, correct.
Response: We apologize for the mistake and it has been corrected.
Other mistakes were corrected. (The title of supporting information file and the 1H and 13C NMR spectra were included in supporting information).